



# 1 A long-term time series of global and diffuse photosynthetically
# 2 active radiation in the Mediterranean: interannual variability and
# 3 cloud effects

Pamela Trisolino[1,2], Alcide di Sarra[1], Fabrizio Anello[3], Carlo Bommarito[3], Tatiana Di Iorio[1], Daniela
Meloni[1], Francesco Monteleone[3], Giandomenico Pace[1], Salvatore Piacentino[3], and Damiano Sferlazzo[4].
[1] ENEA, Laboratory for Observations and Analyses of Earth and Climate, Roma, Via Anguillarese 301, 00123, Italy
[2] Department of Ecological and Biological Sciences, University of Tuscia, Viterbo, Largo dell'Università snc, 01100, Italy
[3] ENEA, Laboratory for Observations and Analyses of Earth and Climate, Palermo, via Principe di Granatelli 24, 90139
Italy.
[4] ENEA, Laboratory for Observations and Analyses of Earth and Climate, Lampedusa, Contrada Capo Grecale, 92010, Italy
Correspondence to: *pamela.trisolino@gmail.com*
**Abstract.** Measurements of global and diffuse photosynthetically active radiation (PAR) have been carried out on the island
of Lampedusa, in the central Mediterranean Sea, since 2002. PAR is derived from observations made with multi filter
rotating shadowband radiometers (MFRSRs) by comparison with a freshly calibrated PAR sensor and by relying on the on-
site Langley plots. In this way, a long-term calibrated record covering the period 2002-2016 is obtained and is presented in
this work.
The monthly mean global PAR peaks in June, with about 160 W m$^{-2}$, while the diffuse PAR reaches 60 W m$^{-2}$ in some cases
in spring or summer. The global PAR displays a clear annual cycle with a semi amplitude of about 52 W m$^{-2}$. The diffuse
PAR annual cycle has a semi amplitude of about 12 W m$^{-2}$ (about 23 % of the annual mean value). The diffuse PAR is about
39 % of the global, with a marked seasonal variation, between about 25-30 % in winter and about 50 % in summer.
A simple method to retrieve the cloud-free PAR global and diffuse irradiances in days characterized by partly cloudy
conditions has been implemented and applied to the dataset. This method allows to retrieve the cloud-free evolution of PAR,
and to calculate the cloud radiative effect. CRE, for downwelling PAR. The cloud-free monthly mean global PAR reaches
175 W m$^{-2}$ in summer, and the diffuse PAR about 40 W m$^{-2}$. The annual semi amplitudes are similar for all-sky and cloud-
free data. The diffuse PAR for the cloud-free cases is about 24 % of the global.
The cloud radiative effect, CRE, on global and diffuse PAR is calculated as the difference between all-sky and cloud-free
measurements. The average CRE is about -14.7 W m$^{-2}$ for the global, and +8.1 W m$^{-2}$ for the diffuse PAR. The smallest
CRE is observed in July, due to the high cloud-free conditions frequency. Maxima (negative for the global, and positive for
the diffuse component) occur in March-April and in October, due to the combination of elevated PAR irradiances and high
occurrence of cloudy conditions. Largest monthly mean values of CRE are at about -31 W m$^{-2}$ for the global (April 2007),



and +18 W m$^{-2}$ for the diffuse component (April 2010).  Summer clouds appear to be characterized by a low frequency of
occurrence, low altitude, and low optical thickness, possibly linked to the peculiar marine boundary layer structure.  These
properties also contribute to produce small radiative effects on PAR in summer.
The cloud radiative effect has been de-seasonalized to remove the influence of annual irradiance variations.  The monthly
mean normalized CRE for global PAR shows a statistically significant high correlation with monthly cloud fraction, cloud
top pressure, and cloud optical thickness, as determined from satellite MODIS observations.  The normalized CRE for
diffuse PAR show lower correlations, although still statistically significant, with cloud fraction and cloud top pressure, while
displays a limited correlation with cloud optical thickness.  Cloud fraction seems to be the most relevant parameter driving
the cloud radiative effects.
Normalized CRE data have been de-seasonalized and related with variations of the de-seasonalized PAR.  A statistically
significant correlation is found between the de-seasonalized PAR and the de-seasonalized normalized CRE.  This correlation
is seasonally dependent, and suggests that about 77 % of the global PAR interannual variability may be ascribed to clouds
variability in winter.
**1 Introduction**
The solar radiation comprised between 400 and 700 nm is defined photosynthetically active radiation (PAR, McCree, 1972),
because it is involved in primary production and photosynthetic processes.  The knowledge of PAR reaching the Earth's
surface is essential for the determination of biomass production and for the understanding of terrestrial and marine
vegetation physiology.
The spectral distribution of solar radiation and PAR, as well as the fraction that reaches the surface, is modified by the
atmospheric components through absorption and scattering by clouds and aerosols, and through absorption by ozone and
other minor gases, as a function of the solar zenith angle, and surface albedo.  Beside solar zenith angle, clouds are the main
modulators of PAR.
Terrestrial and marine vegetation may respond in complex ways to PAR, due to combination of photosynthesis, photo-
inhibition, and photo-damage (e.g., Dimier et al., 2009).  The repartition between direct and diffuse PAR, which depends on
the solar zenith angle, atmospheric properties, and albedo, also affects primary production (e.g., Gu et al., 2002; Mercado et
al., 2009, Min, 2005).  The determination of these two components is very important for the understanding and description
of photosynthetic processes.
Despite its importance, few direct measurements of PAR are carried out routinely, especially for the diffuse component.
These measurements are particularly lacking over the sea, where very few measurements are available.  This applies also to
the Mediterranean, which is characterized by high levels of solar (e.g., Hatzianastassiou et al., 2005) and photosynthetically
active radiation (e.g., Hader et al., 2008).  Due to the complexity of the basin and of the occurring interactions, and the high
anthropogenic pressure, long-term measurement programs have been started in the Mediterranean with the aim of



understanding the regional climate, underlying mechanisms, and impacts. Basin-wide experiments like the Chemistry and
Aerosol Mediterranean Experiment (dedicated to chemistry-climate interactions; Mallet et al., 2016), the Marine Ecosystem
Response in the Mediterranean Experiment (addressing the marine ecosystem; the Mermex group, 2011), and the Hydrology
Cycle in Mediterranean Experiment (dedicated to the water cycle, Drobinski et al., 2014), were started. Long-term
observations play an important role within these experiments, for the understanding of variability and of the definition of
conditions occurring during intensive measurement campaigns. This study is dedicated at discussing the long-term
behaviour of PAR in the central Mediterranean based on observations made at the Atmospheric Observatory on the island of
Lampedusa, in the central Mediterranean. Lampedusa is one of the long-term supersites of ChArMEx (Mallet et al., 2016),
and an atmospheric site of HYMEX. Moreover, the determination of PAR in the marine environment is crucial for the
quantification of the biomass production and the understanding of ecosystem processes. Thus, this study provides basic
information related with different aspects of the regional Mediterranean climate.
Different approaches have been used to derive PAR: measurements with dedicated sensors (e.g., Alados and Alados-
Arboledas, 1999), estimates from satellite observations (e.g., Frouin and Pinker, 1995), as a fraction of the broadband solar
irradiance (e.g., Meek et al., 1984; Alados et al., 1996), or using empirical expressions (Udo and Aro, 2000). In a previous
work (Trisolino et al., 2016), we developed a simple method to obtain calibrated global and diffuse PAR irradiances from
measurements made with a multi-filter rotating shadowband radiometer, MFRSR. The method relies on narrowband
measurements of global and diffuse irradiance in 4 bands within the PAR spectral interval, and on the possibility to
frequently obtain on-site calibrations with the Langley plot method, without interruptions of the routine measurements.
In this work, we apply the method by Trisolino et al. (2016) to MFRSR measurements made at Lampedusa and derive a
long-term series of global and diffuse PAR. The dataset covers the period 2002-2016 and is used to investigate the
behaviour and variability of global and diffuse PAR, and to quantify the influence of clouds on these components. Section 2
describes the site and the instruments used in this study. The long-term time series of global and diffuse PAR is presented in
section 3. Estimates of cloud-free PAR are derived and discussed in section 4, while cloud effects are discussed in section 5.
**2 Site and instruments**
The instruments used for this study are Li-190 PAR sensors and MFRSR radiometers installed at the ENEA (Italian National
Agency for New Technologies, Energy, and Sustainable Economic Development) Atmospheric Observatory on the island of
Lampedusa (35.5° N, 12.6° E; http://www.lampedusa.enea.it/), in the central Mediterranean Sea. The Observatory is located
along the North-Eastern promontory of Lampedusa, at about 45 m above mean sea level. The instruments are installed on
the roof of the main Observatory building, at about 15 m from the Northern cliff of the island; the instruments field of view
is almost totally devoid of obstacles. Lampedusa is a small island (about 20 km² surface area) relatively far (> 100 km) from
continental regions and with a very limited impact from local sources. A large set of instruments for the characterization of
regional climate and relevant parameters (radiation, greenhouse gases, aerosol, water vapour, clouds, ozone, etc.; see, e.g.,



Ciardini et al., 2016) is operational at the Lampedusa Atmospheric Observatory, which contributes to the World
Meteorological Organization Regional Global Atmosphere Watch Network.
Measurements of PAR global irradiance were started in Lampedusa with a Li-Cor 190 radiometer in June 2004.
The PAR sensor consists of a diffuser, a visible bandpass interference filter, and a Si photodiode, and measures
the down-welling PAR irradiance.  According to Ross and Sulev (2000) the systematic spectral errors of the Li-
190 do not exceed 1 %; this sensor also display a good angular response, with a maximum deviation from the
ideal cosine response of 7 % at 80°, and larger deviations for larger angles (Akitsu et al., 2017).  Different Li-190
PAR sensors have been operational at Lampedusa; only data from two freshly calibrated instruments in two
periods are used in this study, essentially to derive regression parameters which allow to obtain PAR irradiances
from MFRSR signals, as in Trisolino et al. (2016).  Two hundred Watt quartz tungsten halogen lamps traceable to
the U.S. National Institute of Standards and Technology (NIST) have been used during the calibration by the
manufacturer. The estimated uncertainty is less than ±5 % (typically ±3 %).  The PAR irradiance measured with
a Li-190, $PAR_L$, is used as reference determination during a period of 6-7 months following the factory
calibration.
The Multi-filter Rotating Shadowband Radiometer uses six narrow-band and one broadband channels with a single input
diffuser and different interference-filter and photodiode detectors, to measure global and diffuse components of the solar
irradiance. The six narrow-bands have a bandwidth (FWHM) of about 10 nm and are centred at the wavelengths of about
415, 500, 615, 673, 879, and 940 nm. With an automatic rotating shadow-band, the instruments measures the diffuse
component of the irradiance. This allows to derive the irradiance direct component as the difference between global and
diffuse measurements (Harrison, 1994).  MFRSR measurements at Lampedusa were started in 2001, and different
instruments have been used in different periods.  The data used in this study are from MFRSR #424, operational from 2001
to 2011, and from MFRSR #586, which was installed in 2011 and has been operational since then.  The central wavelengths
of the 4 bands within the PAR spectral range for the two MFRSR radiometers are reported in Table 1.  MFRSR data are
calibrated on site with the Langley plot method, and the aerosol optical depth is derived routinely at several wavelengths.
Further details on the MFRSR radiometers operating at Lampedusa are given by di Sarra et al. (2015).  In addition to
MFRSR, a Cimel Sun-photometer which is part of the Aerosol Robotic Network (AERONET, Holben et al., 1998) is
operational at Lampedusa.  The MFRSR data are corrected for the aerosol forward scattering for cases of large particles, and
an aerosol optical depth record constituted by the integrated dataset of MFRSR and AERONET data is obtained (di Sarra et
al., 2015).
A calibration of the PAR sensors at least every two years is recommended by the manufacturer. Using the method by
Trisolino et al. (2016) it is possible derive global and diffuse PAR irradiances with a good accuracy, using MFRSR
measurements and taking advantage of the MFRSR in situ calibration with the Langley plot method.  Two sets of regression

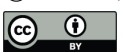



coefficients needed to derive PAR from MFRSR measurements in the 4 bands falling in the PAR range were derived. The
first set was derived as in Trisolino et al. (2016), in the period 2 March-30 May, 2007 by comparing MFRSR #424 signals
with the irradiance measured with a freshly calibrated PAR, # Q36970. The second sets was derived for MFRSR #586, by
comparing its signals with PAR # Q100313 measurements in the period 19 September-21 December 2015. It must be noted
that two different sets of coefficients are needed, due to the somewhat different central wavelengths of the filters of the two
instruments. Li-190 # Q36970 was calibrated at the factory in November 2006, and Li-190 # Q100313 in May 2015.
Following Trisolino et al. (2016), the PAR irradiance was derived by linearly combining the calibrated MFRSR signals in
the 4 channels falling in the PAR spectral range. The following relationship is used to infer calibrated PAR measurements:
$PAR = c_1 S_1 + c_2 S_2 + c_3 S_3 + c_4 S_4,$ (1)
in which $S_1$, $S_2$, $S_3$, and $S_4$ are the signals at channels centred at about 415, 500, 615, and 675 nm, and $c_1$, $c_2$, $c_3$, and $c_4$ are
coefficients. The coefficients are obtained by minimizing the difference between PAR and $PAR_L$. The two sets of
coefficients for the two MFRSR radiometers used in this study are reported in table 2.
As discussed in Trisolino et al. (2016), the estimated uncertainty on the MFRSR-derived PAR is about 5 % for the global,
and 9 % for the diffuse irradiance.
MFRSR-derived diffuse PAR irradiances were compared with radiative transfer model (MODTRAN 5.3) calculations in
Trisolino et al. (2016), because direct measurements of diffuse PAR were not available. A Li-190 sensor, #101552, was
installed on a sun-tracker for diffuse PAR measurements at Lampedusa on 18 February 2016, and has been operational since
then. PAR #101552 was calibrated at the factory in December 2015. Thus, we have the opportunity to verify the estimated
values of diffuse PAR against diffuse $PAR_L$. Overlapping data between 19 February to 25 May 2016 are used in the
comparison. The scatterplot and the fit between 5-minute averages of diffuse PAR derived from MFRSR and diffuse $PAR_L$
are shown in figure 1 for all atmospheric conditions.
The root mean squared difference between the two datasets is around 9 %. If we sum quadratically the absolute uncertainty
on PAR measurement (3-5 %), we have a total estimated uncertainty of about 10 %. This value is in agreement with the
uncertainty estimate based on the comparison with the radiative transfer model reported by Trisolino et al. (2016).
By comparing $PAR_L$ from different Li-190 sensors obtained with the original factory calibration with PAR inferred from
MFRSR measurements, it is possible to determine the long-term variation of the Li-190 sensitivity. All different PAR
sensors display a long-term decrease of the instrumental sensitivity. This decrease, calculated over at least 2 years, is
between 0.6 and 2.3 %/year. This is consistent with the results by Mizoguchi et al. (2010), who found a degradation by 1-3
% after 1 year of field operation for different PAR sensors. Our data show in some cases a faster decline, up to 15 %/year,
for specific sensors over shorter periods (few months), which appears to take place preferably during the first months of use.
Similar reductions were observed by Blum and Schwank (1985) and, more recently, by Akitsu et al. (2017). This behaviour
suggests that a frequent calibration and redundancy in the measurements may be required to obtain accurate PAR
observations. The possibility to rely on on-site routine calibrations, as is the case for the method used in this study,
guarantees a better long-term accuracy than obtainable from standard PAR observations.




## 3 All sky global and diffuse PAR irradiances

The time series of monthly mean global and diffuse component of photosynthetically active radiation from MFRSR
measurements over the period 2002-2016 is shown in fig. 2. The MFRSR data are acquired every 15 s; hourly, and then
daily averages are calculated from the 15 s measurements. Monthly means are calculated from daily values for those months
with at least 27 daily average values. Data acquired at solar zenith angles larger than 80°, affected by a significant cosine
response error (e.g., Mizoguchi et al., 2010; Akitsu et al., 2017), are excluded. This limitation on the solar zenith angle
produces an underestimate of the daily PAR by about $1.4 \pm 0.5$ Wm$^{-2}$ for the global component, and $1.1 \pm 0.3$ Wm$^{-2}$ for the
diffuse component.
Figure 2 shows the monthly mean global and diffuse PAR as derived from MFRSR observations. The two data series
display a typical annual cycle with a summer maximum, driven by the annual course of the solar zenith angle, with
significant interannual variability. This interannual variability is larger for the diffuse than for the global PAR.

### 3.1 Mean annual cycles

Figure 3 shows the annual evolution of monthly median, percentiles, maxima and minima of global and diffuse PAR.
Global PAR shows an evident annual cycle with single summer maximum, while diffuse PAR displays a more articulated
seasonal evolution, with a first maximum in April and a secondary in June.
Alados et al. (1996) show the annual evolution of global PAR in Almeria (36.83° N, 2.41° W), at a latitude close to that of
Lampedusa, but on land. The differences between the monthly median values of the two datasets are generally within 10 %.
Largest differences are found in March.
Yu et al. (2015) show ground based measurements of daily global PAR at several sites of the Surface Radiation, SURFRAD,
network in United States for year 2011. Goodwin Creek (34.3° N, 89.9° W, 98 m a.m.s.l.) and Desert Rock (36.6° N, 116.0°
W, 1007 m a.m.s.l.) are located in the same latitude band of Lampedusa. Despite they represent very different environments,
PAR measurements are in reasonable agreement with those of Lampedusa: differences with respect to Lampedusa are about
3 % in summer and 4 % in winter for Desert Rock, and 12 % in summer and 3 % in winter for Goodwin Creek.
Jacovides et al. (2004) report global PAR observations made over three years (September 1997-May 2000) in Athalassa,
Cyprus, in the Eastern Mediterranean. The daily values of PAR in Cyprus are in good agreement with those at Lampedusa,
with differences between maximum mean values of about 4 % in summer and 8 % in winter.
Very few studies are dedicated to the annual evolution of the diffuse PAR and, to our knowledge, no direct measurements
over the sea or on small islands are available.
Measurements of diffuse PAR at latitudes similar to that of Lampedusa were reported by Dye (2004) and Wang et al. (2017).
Dye et al. (2004) shows measurements of diffuse PAR at the Southern Great Plains Atmospheric Radiation Measurement,

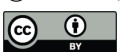



ARM, program site (36.6° N, 97.5° W; 314 m) between 1 March and 31 July 2000. Differences between the monthly mean
diffuse PAR at Southern Great Plains and Lampedusa are between 7 % in July to 19 % in June.
Wang et al. (2017) applied various methods to estimate the diffuse PAR to observations made in Canada and in USA with
Detlta-T BF3 sensors.  Two of the sites used in this study are located at the same latitude of Lampedusa (US-fuf, 35.09° N,
111.76° W, and US-fwf, 35.45° N, 111.77° W) but are at about 2200 m above mean sea level (2180 m and 2270 m,
respectively) and in different terrestrial environments (woody savannas and grassland, respectively). The monthly diffuse
PAR at Lampedusa is 25-54 % of that at US-fwf, and 17-52 % of that at US-fuf.  This large difference is primarily due to
altitude and albedo, and possibly clouds.
Differences in altitude, surface properties and albedo, aerosol, and clouds are expected to play a large role in the diffuse PAR
modulation, and the agreement found between sites at similar latitude may be somewhat incidental.
After solar zenith angle, clouds are the main modulation factor, which are expected to act differently on the global and
diffuse PAR components.
In general, if we exclude global irradiance enhancements which may occur with scattered cloud conditions, the global
irradiance for cloudy conditions is expected to be always lower than for cloud-free sky.  Conversely, the diffuse irradiance is
larger than in cloud-free conditions for thin and moderate clouds, and is smaller for thick clouds.  Thus, the relationship
between cloudiness and diffuse PAR is expected to be more complex than for the global component.
Figure 4 shows the monthly mean occurrence of cloud free conditions (cloud cover approximately lower than 1 okta and
direct radiation unobstructed; see section 4.1 for more details on the cloud-free identification scheme) between 9 and 14
UTC.  The monthly median and quartiles are calculated over the 2002-2016 time interval, considering only months with at
least 27 days with measurements.
The largest interquartile range of the global PAR irradiance is observed to occur in April and August.  This is due to the
combination of elevated PAR levels and a higher cloud occurrence than in June and July.
Largest median values of diffuse PAR are found in April.  The monthly diffuse PAR median is lower in summer than in
April, although the solar zenith angle is higher, due to the higher rate of occurrence of cloud-free conditions.  In addition to
the cloud occurrence, cloud properties are expected to play a significant role in determining the amount of global and diffuse
PAR reaching the surface.
The average diffuse-to-global PAR ratio is 0.39±0.08, and displays a marked seasonal variation, with maxima around 0.5 in
winter and minima at about 0.25-0.3 in summer.
The monthly time series shown in figure 2 was fit with the following simple oscillatory function:
$f(t) = a + b \sin [\omega (t + \varphi )]$      (2)
with the aim of quantitatively determining the main parameters of the PAR evolution.  The coefficients $a$, $b$, and $\varphi$, retrieved
for the global and diffuse PAR are reported in table 3, while the fitting curve is also included in figure 2.  The variable $\omega$ is
$2\pi/12$ and $t$ is in months.





As discussed in section 2, our time series is based on the determination of weighting coefficients for the signals of two
MFRSR instruments against two freshly calibrated PAR sensors.  The two determinations took place in 2007 and 2015.
Unfortunately, we do not have overlapping reliable PAR observations in 2011, when MFRSR #424 stopped working and
was replaced by MFRSR #586.  Thus, the series relies on two independent calibrations of PAR sensors, whose nominal
accuracy is between 3 and 5 %.  Thus, we can not exclude that a step change in calibration, within the nominal accuracy of
the used sensors, took place in 2011.  Since there is no independent verification with a higher degree of accuracy, we have
preferred to avoid calculating trends on the time series.  Thus, we did not include a term describing the trend in $f(t)$.
As expected, the correlation coefficient between data and the fitting function is higher for global than for diffuse PAR.  The
ratio between the annual semi amplitude, $b$, and the long-term average is larger for the global than for the diffuse PAR (0.55
against 0.36).
**4 Cloud-free global and diffuse PAR irradiances**
As discussed above, clouds play a large role in modulating the PAR evolution and variability.  The determination of PAR in
cloud free conditions is necessary to quantitatively assess the role played by clouds.  Thus, we have developed a method to
infer monthly mean cloud-free global and diffuse irradiances, which may be used to determine the cloud radiative effect in
the PAR spectral range.

**4.1 Determination of cloud-free PAR**
The determination of cloud-free PAR irradiances is based on the identification of cloud-free conditions.  Measurements of
MFRSR radiometers are used to select cloud-free periods. The algorithm developed by Biavati et al. (2004) is an adaptation
of the method by Long and Ackerman (2000) to the characteristic conditions and properties of the aerosol observed at
Lampedusa.  This algorithm, briefly described by Meloni et al. (2007), allows to determine periods with no clouds, also with
high values of aerosol optical depth.  The method is very selective and allows identifying sky conditions virtually devoid of
clouds.  A comparison with sky imager pictures shows that the algorithm selects cases with cloud cover ≤1 okta and Sun
unobstructed by clouds.  This method has been used to select conditions suitable for the determination of aerosol radiative
effects, for which the cloud radiative influence needs to be negligible (see, e.g., di Sarra et al., 2008; Di Biagio et al., 2010).
With the aim of estimating the cloud-free irradiance during partly cloudy days we applied to PAR the method proposed by
Long and Ackerman (2000) for the shortwave radiation.  The following function
$F(\theta) = A(cos\theta)^B$                                                                                                       (3)





is fit to the PAR data in cloud-free intervals, and is used to estimate the cloud-free expected irradiance during cloudy
intervals. The variable θ is the solar zenith angle and *A* is the cloud-free irradiance for a solar zenith angle of 0°; the fitting
curve includes the effects of the atmosphere constituents (changes in atmospheric pressure, aerosol, and absorbing gases)
which affect PAR and are assumed to vary slowly, and possible residual instrumental calibration inaccuracies. The *B*
coefficient includes possible effects of the cosine response of the instrument.  The contribution of water vapour to the
modulation of PAR is negligible; this is an advantage in applying this fitting procedure to PAR data with respect to the
whole shortwave range.
Figure 5 shows the measured PAR and the derived fit for a day with only a portion with cloud-free conditions.
The coefficients are determined for every partly cloudy day. The Long and Ackerman method requires some minimum
number of cloud-free measurements over a significant range of solar zenith angle.  The method is not applicable to days with
persistent overcast conditions, or when few short cloud-free intervals are present.  Thus, only a selection of cases is retained
in the cloud-free dataset.  These are selected by visually looking at the fitting curve and eliminating the days with incomplete
or inadequate fits.

**4.2 Long-term cloud-free PAR irradiances**
We obtain a long-term time series of cloud-free PAR estimates using the method described in section 4.1. The monthly mean
values for cloud-free conditions are calculated from daily means with at least 11 daily values.  During the winter months also
averages calculated with at least 5 values are accepted for diffuse PAR, due to the low number of available days for which it
is possible to estimate the cloud-free irradiance.  The monthly mean time series of global and diffuse irradiance is shown in
fig. 2 together with those obtained in all sky conditions.
As expected, the global irradiance is always larger than for all sky conditions, while the opposite occurs for the diffuse PAR.
The average diffuse-to-global PAR ratio is 0.237±0.035; the seasonal variation is much smaller than for all-sky conditions.
As for the PAR measurements in all sky conditions, the time series was fit with expression (1).  The fitting coefficients for
cloud-free series are reported in table 4.
The values of the annual semi-amplitudes are very close for all-sky and cloud-free conditions, while the long-term average is
larger for cloud-free conditions for global PAR, and larger for all-sky conditions for diffuse PAR.  The ratio between the
annual cycle semi-amplitude and the long-term average is about 0.45 for both the global and the diffuse components for
cloud-free conditions.  The ratio between the diffuse and the global PAR long-term averages is 0.24; this is significantly
smaller than for all-sky conditions.



### 4.3 Mean annual cycles

The average annual evolution of global and diffuse cloud-free PAR is shown in fig. 6. The PAR interannual variability in this case is primarily attributable to aerosol. As discussed by di Sarra et al. (2015) and by Di Iorio et al. (2009), the aerosol distribution at Lampedusa is characterized by a significant annual cycle, essentially modulated by the occurrence of desert dust events, which generally occur in spring and summer (Meloni et al., 2007). The day-to-day variability is largest in spring and summer, and smallest in winter. This behaviour is evident in the variability of both global and diffuse PAR. Previous studies carried out at Lampedusa have shown that this aerosol variability largely affects UV (di Sarra et al., 2002; Meloni et al., 2005; Casasanta et al., 2011), solar radiation (di Sarra et al., 2008; Di Biagio et al., 2010) and, for desert dust particles, infrared radiation (di Sarra et al., 2011; Meloni et al., 2015).

Figure 7 shows the comparison between the monthly average for all-sky and for cloud-free conditions. The difference between the two curves is the monthly mean cloud radiative effect, CRE, which is produced by clouds on downwelling global and diffuse PAR at Lampedusa. The reader is reminded that this analysis is based on a selective cloud screening, and partially cloudy or short cloud-free intervals are classified as cloudy by our algorithm.

### 5 Cloud effects

The differences between all-sky and cloud-free global and diffuse downwelling irradiances are shown in figure 8. As expected, CRE is negative (a reduction of down-welling PAR) for global, and positive (an increase) for diffuse PAR. The long-term average cloud radiative effect is about -14.7 W m$^{-2}$ for the global component, and +8.1 W m$^{-2}$ for the diffuse PAR. The interannual variability is significant.

Extended summer periods with small values of CRE (>-7 W m$^{-2}$ for the global component) occur in 2003, 2008, 2009, 2012, 2013, and 2015. It is interesting to note that elevated summer temperatures were recorded at Lampedusa during the summer seasons of 2003, 2009, 2012, and 2015 (Ciardini et al., 2016). Some of these events (e.g., 2003, 2015) were recorded at the regional level.

Extended time intervals with large cloud effects occurred in several periods (early 2003, 2005, 2006, 2010), while cases with large cloud effects over limited periods are also present (late 2007, early 2013 and 2014, mid 2016). Some of these negative peaks appear to correspond with cold winters at Lampedusa (e.g., in 2003, 2005, 2006, 2009, 2012, 2013, and 2015; Ciardini et al., 2016).

Figure 9 shows the annual evolution of the monthly median CRE and its variability, for the global and diffuse component. The absolute value of the diffuse CRE displays an annual evolution similar, but opposite in sign, to the global. The absolute value of CRE is smallest during months characterized by dominant cloud-free conditions, generally in late summer, and shows a maximum generally between March and April, and sometimes a secondary maximum in October. For the diffuse CRE the maximum variability takes place in September, and the largest CRE median in April.



Pyrina et al. (2015) investigated the effect of clouds on the shortwave radiation in the Mediterranean basin in the period
1984-2007 based on satellite observations and radiative transfer model calculations. The seasonal patterns of the surface
CRE in the shortwave, $CRE_{SW}$, is similar to that for CRE of global PAR in fig. 10, with maximum values in April-May and a
secondary maximum in October. The secondary maximum in October is less pronounced in the analysis by Pyrina et al.
than at Lampedusa. Data by Pyrina et al. show a significant variability of $CRE_{SW}$ throughout the basin, and particularly in
the North-South direction, except in October.
Pyrina et al. (2015) also show that the radiative effects depend on the cloud type.
We have investigated this aspect by extracting monthly average cloud properties retrieved by the Moderate Resolution
Imaging Spectroradiometer, MODIS, on board Aqua over a region 2°x2° around Lampedusa. Level 3 monthly mean Cloud
Optical Thickness (COT), Cloud Fraction (CF) and Cloud Top Pressure (CTP) determined from July 2002 to December
2016 are used (MOD08_M3_6). Monthly averages from daytime observations from MODIS Aqua, whose daytime passage
over Lampedusa is close to solar noon, have been selected.
Level 3 MODIS products are based on a cloud mask for the identification of cloudy pixels, with a resolution of 5 km.
Cloud Optical Thickness is derived from multispectral reflectance measurements compared with parametrizations obtained
from a theoretical model, simulated in function of 3 different geometrical angles. The look-up table of the theoretical model
results contains reflectance values, effective radii, and surface albedo (King et al., 1997).
The cloud fraction is calculated as the ratio between the number of cloudy pixels and the total number of pixels in the image.
CTP is determined using the radiance measurements in the $CO_2$ absorbing region from 13.3 to 14.2 μm using a $CO_2$-slicing
technique. An infrared channel at 11 μm is also used for optically thick and lower-level clouds (Platnick et al., 2015).
Figure 10 shows the annual evolution and variability of monthly values of COT, CF, and CTP. A significant co-variance, in
particular of CF and CTP, is apparent, with lowest clouds and smallest CF in July, when also a minimum of COT is
observed. Thus, cloud properties vary seasonally, with lower, although less frequents and thinner, clouds occurring in
summer. The annual evolution of CF and of the fraction of cloud-free conditions of fig. 4 are also highly consistent.
Contrarily to expectations, low level clouds are associated, especially in July, with low values of COT. This may be related,
in addition to prevailing synoptic conditions, to the seasonal evolution of the planetary boundary layer at Lampedusa, which
is dominated by the thermal difference between the sea surface temperature and the atmosphere. This produces strongest
temperature inversions during summer (see e.g., Pace et al., 2012), with the possible formation of thin low level clouds at the
top of the inversion layer, which is usually located within 200 m of the sea surface. The frequency of occurrence and
properties of clouds in summer favour high level of PAR and, similarly, UV radiation (e.g., Meloni et al., 2005). As
discussed by Becagli et al. (2013), these elevated radiation levels, together with a shallow marine mixed layer, stress the
marine biological cells; these mechanisms lead to increased di-methyl sulphide emissions and high values of he
phytoplankton productivity index (rate of photosynthetic carbon assimilation per weight of chlorophyll a) in summer, while
the chlorophyll amount displays a maximum in winter.





The largest cloud radiative effects occurring in the transition seasons appear to be associated with moderate values of COT
(except in November, when the median is higher and the minimum-to-maximum COT excursion is largest) and relatively
high clouds.
CRE is modulated, in addition to clouds, by the annual evolution of the photosynthetic irradiance. To reduce the effect of
the seasonal changes in irradiation we have defined as normalized cloud radiative effect, NCRE, the ratio between CRE and
the cloud-free PAR irradiance of the corresponding month. NCRE is between about -0.5 and 0 for the global PAR, and
between 0 and +0.8 for the diffuse PAR, with larger relative variations for diffuse than for global PAR.
Figure 11 shows the time series of global and diffuse NCRE together with cloud fraction, cloud optical thickness, and cloud
top pressure determined by MODIS. The variability of cloud properties helps explaining the interannual variability of
NCRE, in particular for global PAR.
Winters 2009-2010 and 2015-2016 are characterized by a less pronounced seasonal minimum of global NCRE, which is
associated with low values of COT in these seasons compared to the other years. The winter 2009-2010 clouds produce also
the largest effect on the diffuse NCRE. The COT seasonal cycle appears to be attenuated 2007, with an anticipation of the
winter maximum already in 2006 and a lacking summer minimum. This seems to produce a smaller winter negative peak of
NCRE, but does not seem to affect the summer maximum, probably because of the annual evolution of CF and the
prevailing role of cloud-free conditions. The largest NCRE for the month of April is also found in 2007. This is also the
largest CRE of the whole record, and appears to be due to elevated values of COT in this season compared with the other
years; CTP and CF do not show significant anomalies in this season.
Several winter seasons display minima of the global NCRE less than -0.3. Two of the three largest NCRE negative peaks, in
early 2006 and 2014, coincide with maxima of both COT and CF. The other peaks, in early 2003, December 2004, early
2008 and 2012, coincide with maxima of one between COT and CF.
Figure 12 shows scatterplots between NCRE and cloud properties derived from MODIS. Cross correlation coefficients
between NCRE (global and diffuse) and the different cloud properties were calculated and are reported in table 5. A
statistically significant correlation is found between all considered cloud properties and global PAR NCRE, with the largest
positive correlation occurring with CF, as also is apparent from figure 12.
Diffuse NCRE displays a statistically significant negative correlation with CTP, and positive correlation with CF. As
expected, due to the mixed influence of cloud optical thickness on diffuse PAR, a limited correlation is found between COT
and diffuse CRE. A decrease of NCRE seems apparent for values of COT larger than about 15. The low values of NCRE
for diffuse PAR for low level clouds confirm the role played by summer thin clouds.
Thus, cloud properties may explain a large fraction of the global PAR interannual variability, and may be used to estimate
global PAR from correlative measurements. A weaker relationship relates cloud properties and diffuse PAR, which appears
more difficult to be estimated based on correlative measurements. This confirms that direct measurements of diffuse PAR
are very critical for the understanding of photosynthetic and climate-related effects.





The monthly mean time series was rearranged in order to remove the seasonality and better investigate the interannual
variability. The long-term monthly mean annual cycle was calculated from the datasets, as the average of all values for each
month (all values for January, then for February and so on) of the record. The de-seasonalized time series is thus calculated
as:
$P_{ij} = \frac{PAR_{ij} - \overline{PAR}_i}{\overline{PAR}_i}$                          (4)
where $PAR_i$ is the long-term monthly mean for month $i$ ($i=1, .., 12$), and $PAR_{ij}$ is the monthly mean value for month $i$ and
year $j$. $P_{ij}$ is thus the fractional change in global PAR with respect to its long-term monthly mean. Similarly, a
deseasonalized value of NCRE, $N_{ij}$, was calculated.
Figure 13 shows the scatterplot between de-seasonalized PAR and NCRE. The variations of $P_{ij}$ are within ±20 %, while $N_{ij}$
varies within about ±80 %. The best linear fit is also shown in figure 13. The correlation coefficient is relatively low when
all data are included in the scatterplot. Data were also sorted according to the season, over an extended winter (months of
November, December, January, February, and March) and an extended summer (months of May, June, July, August, and
September) period, and the corresponding scatter plots are included in figure 13. A much stronger correlation is found for
the winter months, when NCRE is largest and appears to explain about 77 % of the PAR interannual variability. The slope
of the $P_{ij}$ versus $N_{ij}$ fitting line in summer is 4 times smaller than in winter; the range of $P_{ij}$ values is within ±0.2 in winter,
and ±0.1 in summer. The correlation is much weaker in summer, with clouds explaining only 31 % of the PAR interannual
variability. This is attributed to the high occurrence of cloud free conditions, and the larger role of atmospheric aerosols,
which typically display a maximum in summer.
**6 Summary and conclusions**
This analysis is based on a new method to estimate global and diffuse component of PAR from calibrated MFRSR
measurements (Trisolino et al., 2016). The method is applied to the long-term MFRSR record at the ENEA Atmospheric
Observatory on the island of Lampedusa. As a complement to the previous study, direct measurements of diffuse PAR with
a Li-190 sensor on a solar tracker were used to verify the reliability of MFRSR diffuse PAR estimates. The estimated
uncertainty is in agreement with previous results obtained from the comparison with radiative transfer model calculations.
Main results of this study may be summarized as follows:
1. the long-term global and diffuse PAR record in the central Mediterranean covers the period 2002-2016. Measurements
are being continued, and the record will be expanded in the future. The long-term mean is 95 and 35 W m$^{-2}$ for global and
diffuse PAR, respectively. The mean annual cycle semi-amplitude is about 55 % and 36 % of the long-term mean for the
global and diffuse PAR, respectively. Interannual variations of global and diffuse PAR appear to be essentially related with
the clouds frequency of occurrence. The cloud occurrence is very low in summer, especially in July.



2. The daily PAR irradiance for cloud-free conditions was estimated also during partially cloudy days by fitting the cloud-free data with an analytical function which depends on the solar zenith angle. In this way, cloud-free global and diffuse PAR time series were derived. The cloud-free long-term mean global PAR is 110 W m$^{-2}$, about 16 % more than in all-sky conditions. Conversely, the cloud-free diffuse PAR long-term mean PAR is 26 W m$^{-2}$, about 24 % less than in all-sky conditions. The cloud-free annual cycle semi-amplitude is 45 % for global PAR, and 47 % for diffuse PAR. The differences in the annual semi-amplitude-to-long-term mean ratio with respect to the all-sky conditions are essentially driven by changes in the long-term mean values.

3. By using all-sky and cloud-free PAR data we have calculated the cloud radiative effect, CRE, for global and diffuse PAR. Global and diffuse CRE display a similar annual evolution, but opposite sign, positive for diffuse and negative for global PAR. The long-term mean CRE is -14.7 W m$^{-2}$ for global PAR and +8.1 W m$^{-2}$ for diffuse PAR. The CRE seasonal evolution is characterized by a primary maximum in late winter/early spring, and a secondary maximum in autumn. The largest global PAR monthly CRE is -31 W m$^{-2}$ in April 2007, while the smallest is generally observed in July, in correspondence with the lowest occurrence of clouds. The largest diffuse PAR CRE is recorded in April 2010, with +18 W m$^{-2}$.

4. The CRE has been scaled by taking into account the monthly cloud-free estimated PAR to remove the influence of the solar zenith angle annual course. The normalized CRE has been associated with cloud properties (fraction, optical thickness, and top pressure) as derived from space with MODIS. The deseasonalized NCRE displays a statistically significant high correlation with the cloud properties (for global PAR, positive with respect cloud top pressure, and negative with respect to cloud fraction and optical thickness), and in particular with cloud fraction. The correlation between diffuse PAR NCRE and cloud properties is weaker than for global PAR. The cloud influence on diffuse PAR is more difficult to disentangle, due to the stronger dependency on cloud properties. Clouds are expected to produce an increase or a decrease of diffuse PAR depending on their optical thickness.

5. De-seasonalized global PAR data were related with de-seasonalized NCRE. The correlation analysis between these two parameters suggests that cloud properties explain about 77 % of the interannual global PAR variability in winter, and about 31 % in summer.

6. The large values of global PAR found in summer appear to be due to the combination of low cloud frequency of occurrence and low cloud optical thickness. The summer clouds are also characterized by a high top pressure, suggesting that low level thin clouds, possibly forming at the top of the summer temperature inversion, play a large role. Also due to these cloud properties, high PAR levels are reached in summer. The high PAR and UV radiation has been observed to stress marine phytoplankton in this region of the Mediterranean (Becagli et al., 2013).

This study contributes to provide new information about the behaviour of global and diffuse PAR in the Mediterranean, and in the marine environment. Information on diffuse PAR is particularly relevant and difficult to be found in the literature.

As a prosecution of this study, we plan to better investigate the variability of cloud-free PAR, in particular with respect to the role of atmospheric aerosol, which play an important role in the Mediterranean radiative processes.





**Acknowledgements**
This study has been partly supported by the Italian Ministry for University and Research through the NextData and
RITMARE projects. We acknowledge the MODIS mission scientists and associated National Aeronautics and Space
Administration personnel for the production of the data used in this research effort. MODIS data were downloaded from the
Giovanni online data system, developed and maintained by the NASA Goddard Earth Sciences Data and Information
Services Center. This study contributes to ChArMEx WP-4 on Aerosol-Cloud-Radiation-Climate and WP-6 on Variability
and Trends.

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



**Table 1. Nominal central wavelengths (in nm) of the 4 band used in the PAR estimate for the two MFRSR radiometers. The**
**FWHM bandwidth is about 10 nm for all channels.**

| Band number | MFRSR #424 | MFRSR #586 |
| --- | --- | --- |
| 1 | 415.6 | 414.7 |
| 2 | 495.7 | 495.5 |
| 3 | 614.6 | 613.6 |
| 4 | 672.8 | 672.1 |






**Table 2. Coefficients, in W/m$^2$, of the linear fit between PAR and MFRSR calibrated signals for the 4-band linear model.**

| MFRSR s/n | $c_1$ | $c_2$ | $c_3$ | $c_4$ |
|---|---|---|---|---|
| **424** | 96.09 | 2.30 | -28.94 | 271.5 |
| **586** | 240.50 | -218.23 | 91.58 | 252.432 |






**Table 3. Coefficients of the fit, number of data and correlation coefficient for the global and diffuse monthly PAR time series.**

|  | a (W m$^{-2}$) | b (W m$^{-2}$) | φ (months) | n | r$^2$ |
|---|---|---|---|---|---|
| *Global PAR* | 95.2 | 52.4 | -3.3 | 161 | 0.976 |
| *Diffuse PAR* | 34.7 | 12.5 | -3.0 | 161 | 0.798 |





**Table 4. Coefficients of the fit, number of data and correlation coefficient for the cloud-free global and diffuse monthly PAR time series.**

|  | a (W m$^{-2}$) | b (W m$^{-2}$) | φ (months) | n | r$^2$ |
|---|---|---|---|---|---|
| *Global PAR* | 109.9 | -49.2 | -3.1 | 158 | 0.975 |
| *Diffuse PAR* | 26.2 | 12.2 | -3.4 | 131 | 0.816 |





**Table 5. Correlation coefficients between clouds properties and NCRE for global and diffuse PAR. Statistically significant at 95 % confidence level are shown in bold.**

|  | *COT* | *CF* | *CTP* |
|---|---|---|---|
| **NCRE, global PAR** | **-0.668** | **-0.903** | **0.709** |
| **NCRE, diffuse PAR** | 0.461 | **0.647** | **-0.592** |



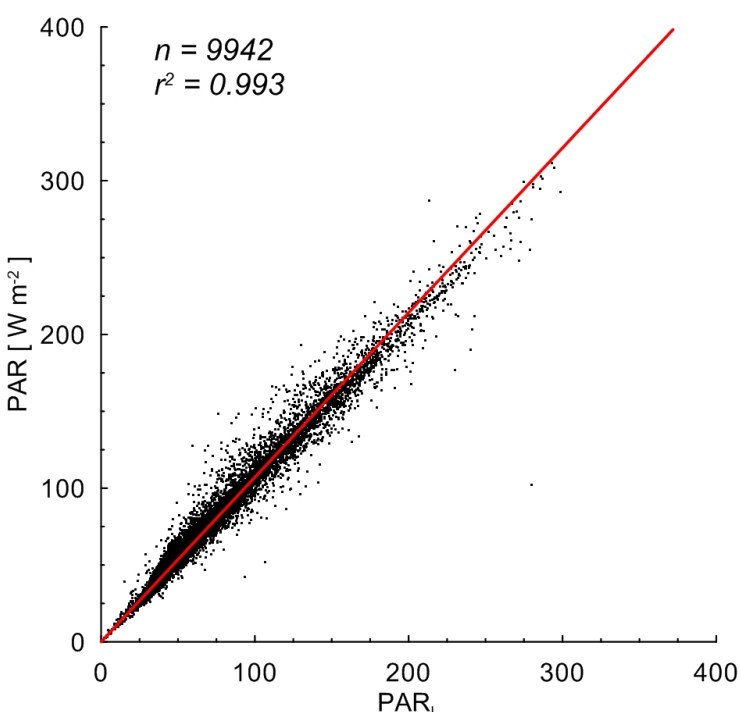

Fig. 1. Scatterplot between 5-minute averages of PAR and PAR$_L$ for the period 19 February-25 May, 2016 at Lampedusa.  The
linear fit is also shown.

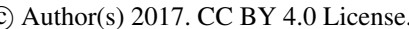



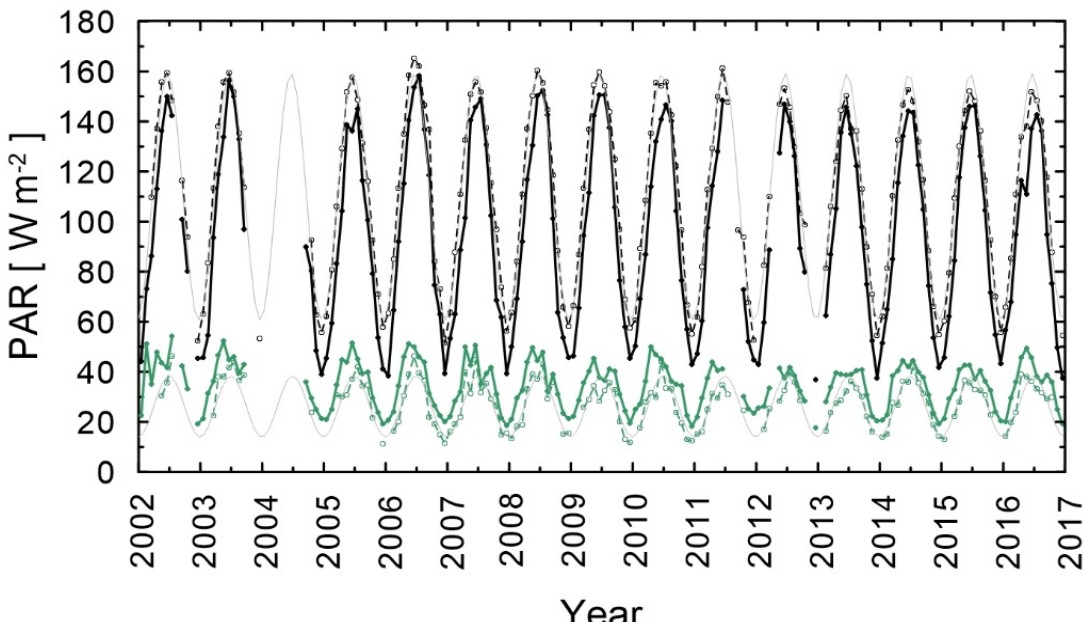

**Fig. 2. Time series of monthly mean global (black lines) and diffuse (green lines) PAR for cloud-free (dashed lines) and all sky**
**conditions (continuous lines). Fitting curves (see text) are shown by thin lines for the cloud-free time series.**






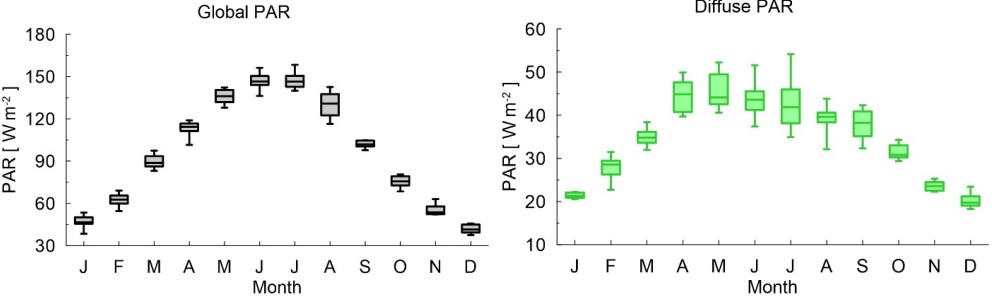


**Fig. 3. Annual evolution of median, lower and upper quartiles, maxima and minima of global (left) and diffuse (right) monthly average PA**


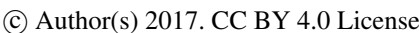




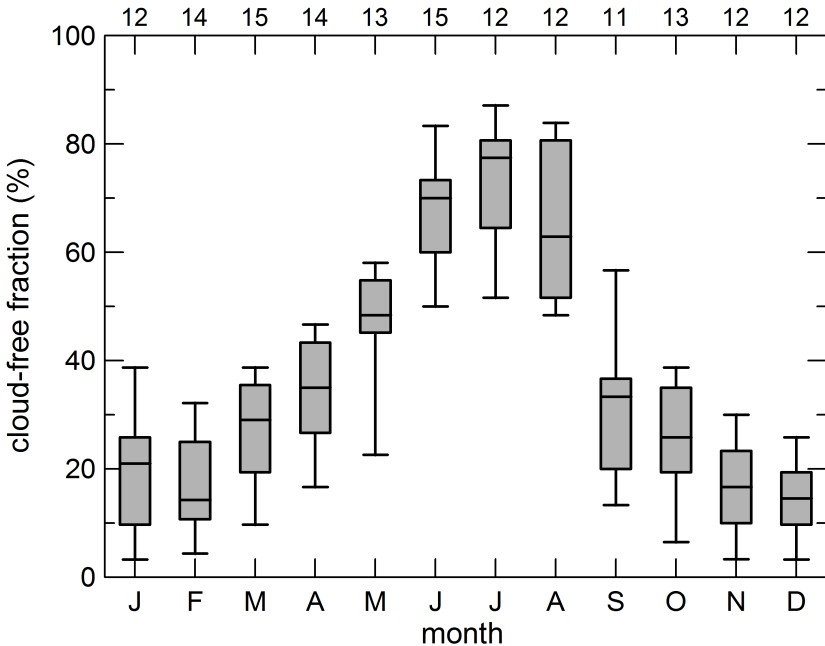

**Fig. 4.** Monthly statistics of the fraction of cloud-free sky between 9 and 14 UTC. Only months with at least 27 days with
measurements in the period 2002-2016 are considered. Minimum, lower quartile, median, upper quartile, and maximum are
indicated (from the bottom to the top) for each month. The number of monthly mean values used for each month is displayed
above the top axis.



**Fig. 5. PAR measured during 4 June, 2016, a partially cloudy day (black line). The cloud-screened data are shown as black dots, and the cloud-free fitting function is shown in red.**





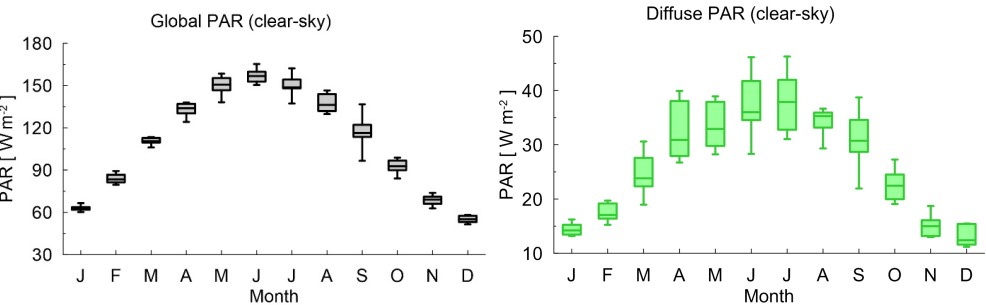

**Fig. 6. Annual evolution of global (left) and diffuse (right) cloud-free PAR; maxima, upper quartile, median, lower quartile, and minima of the monthly mean are displayed.**












**Fig. 7. Global (black) and diffuse (green) monthly average PAR in all-sky (solid lines) and in cloud-free (dashed lines) conditions.**












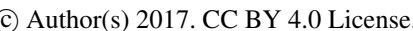



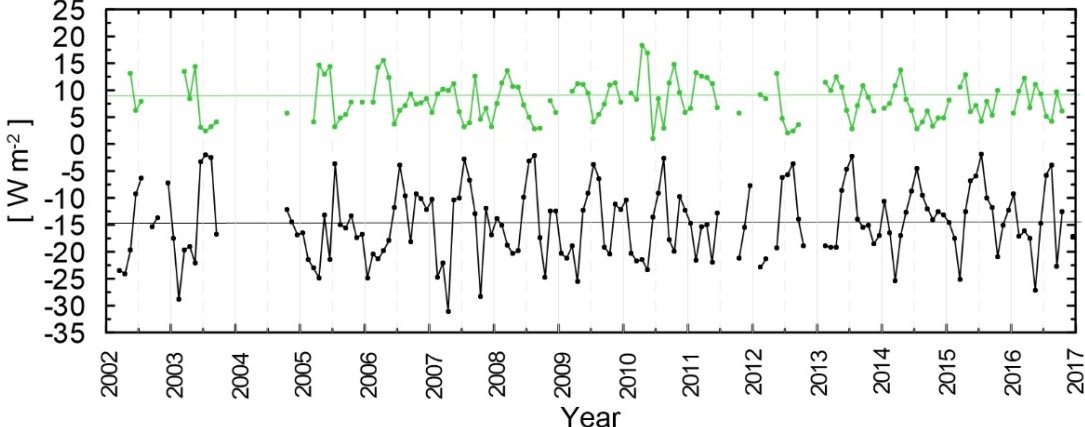


**Fig. 8. Time series of monthly mean cloud radiative effect for global (black) and diffuse (green) PAR, calculated as the difference between all-sky condition and cloud-free monthly mean PAR (see text). Horizontal lines are mean values of global and diffuse PAR, respectively.**







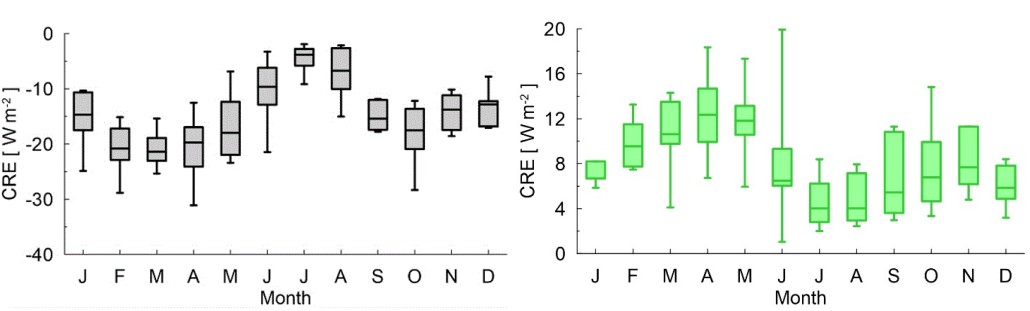


**Fig. 9. Annual evolution of global (left) and diffuse (right) PAR CRE.  Maxima, upper quartile, median, lower quartile, and**
**minima of the monthly values are displayed.**








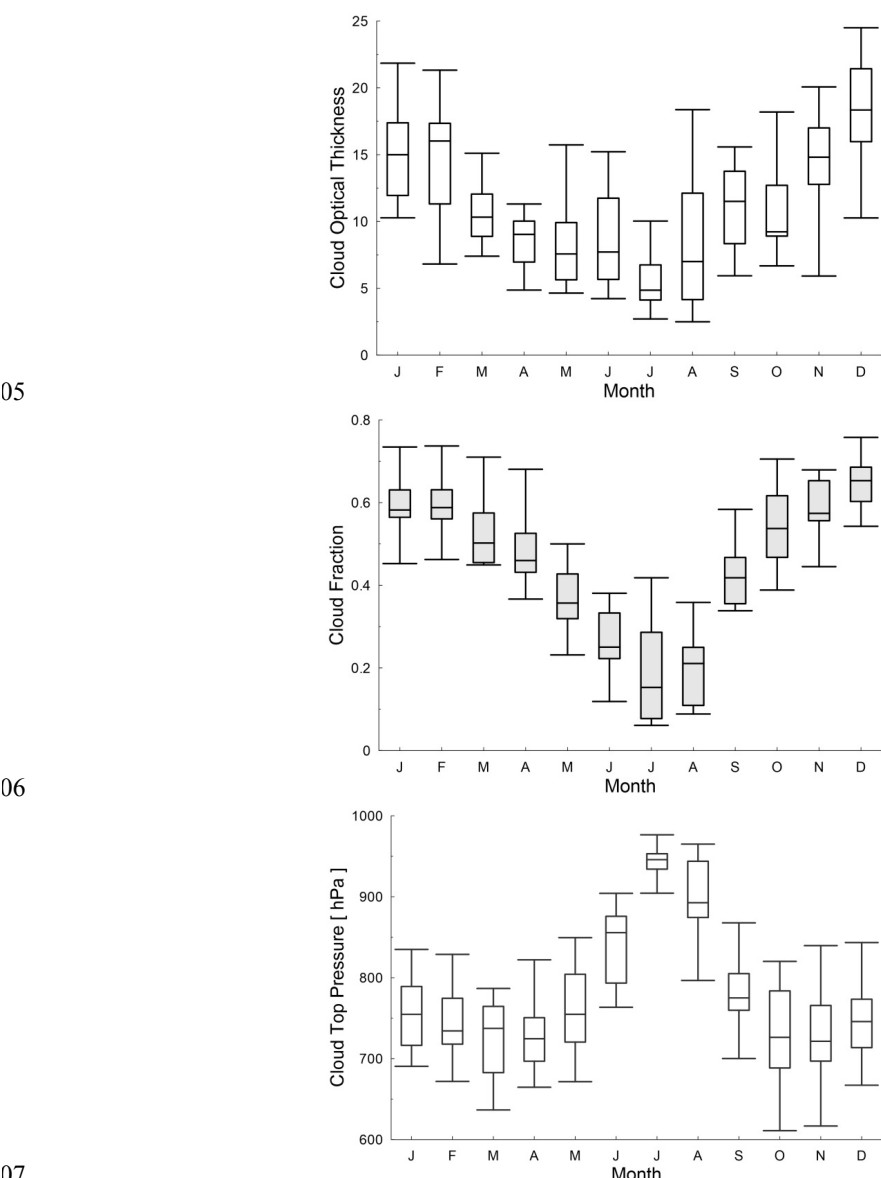

**Fig. 10. Annual evolution of monthly averages of Cloud Optical Thickness (top), Cloud Fraction (middle), and Cloud Top Pressure**
**as derived from MODIS Aqua between July 2002 and December 2016. Maximum, upper quartile, median, lower quartile, and**
**minimum are displayed for each month.**


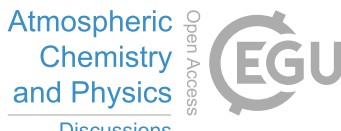

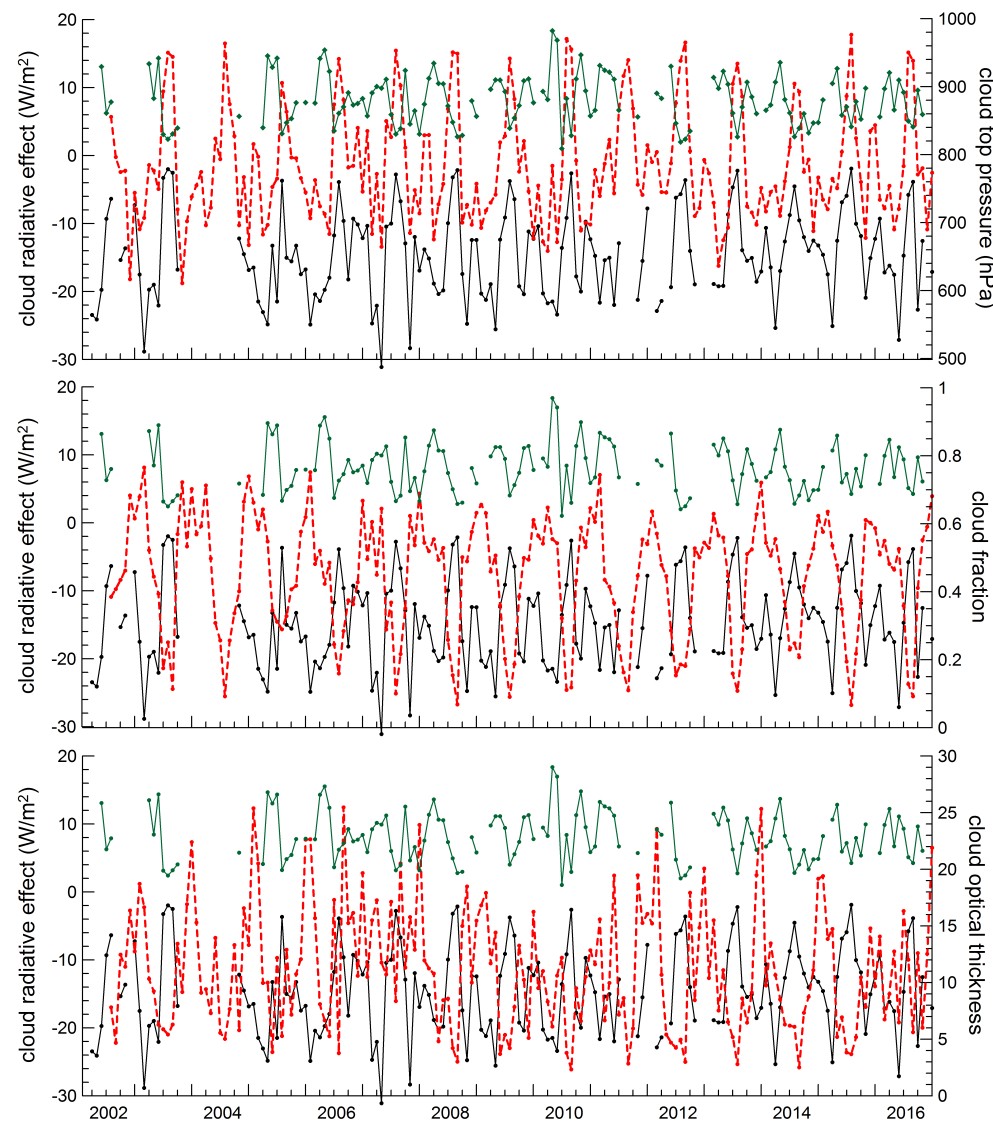

**Fig. 11. Time series of monthly normalized cloud radiative effects for global (black) and diffuse (green) PAR, and cloud top pressure (top graph, red dashed line), cloud fraction (middle, red dashed line) and cloud optical thickness (bottom, red dashed line). Cloud properties are derived from MODIS Aqua observations.**




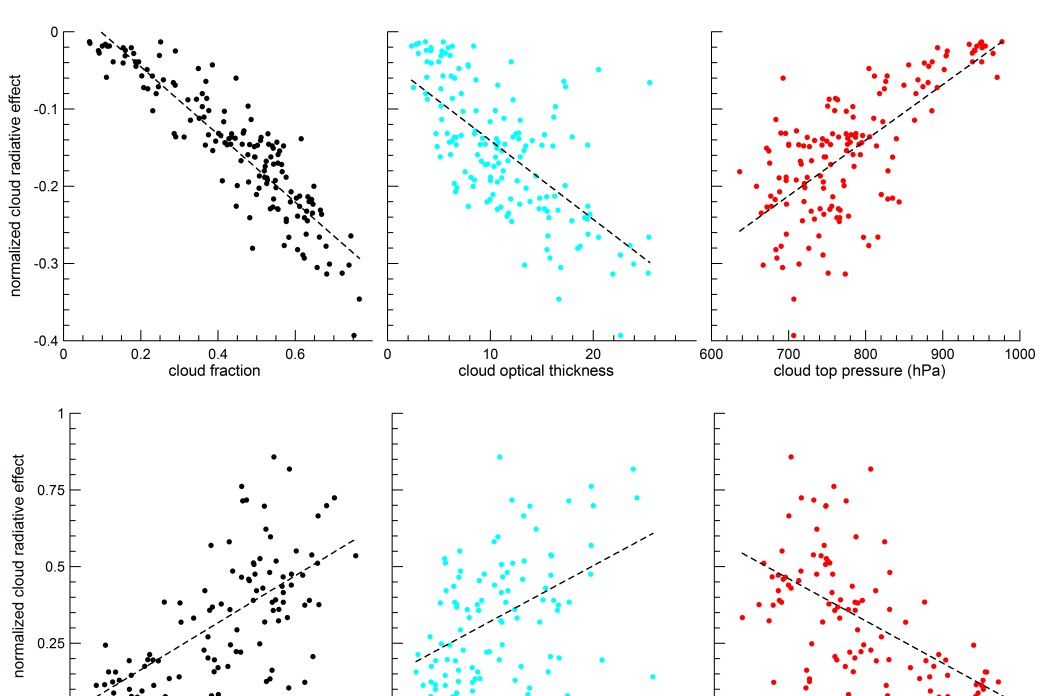


**Fig. 12. Scatterplot between the normalized cloud radiative effect and cloud properties. Graphs for global (top; 151 data points) and diffuse (bottom; 104 data points) PAR are shown. Dashed lines are best linear fits.**








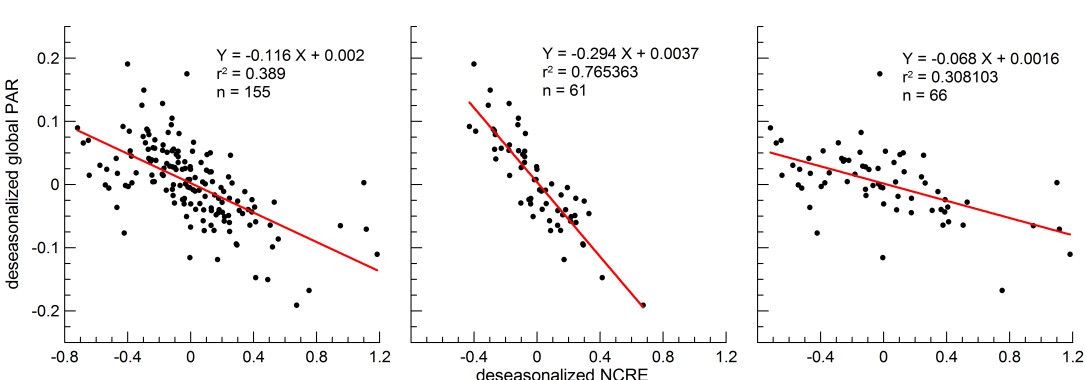


**Fig. 13. Scatterplot between deseasonalized global PAR and deseasonalized NCRE for all data (left), for the months November-**
**March (middle), and for the months May-September (right). Least square fitting lines are plotted in red; the best fit equation,**
**squared correlation coefficients, and number of data points are shown in each graph.**