# Peer review of "active radiation in the Mediterranean: interannual variability and"

_Atmospheric Chemistry and Physics, 2017_

## Referee Comment (RC1) · Anonymous Referee #1 · 27 Dec 2017

This is a very interesting study on PAR measurements.

There are not too many studies on PAR long term measurements and this study is presenting a lot of new and iteresting aspects of global and diffuse par

I would suggest the publication of the manuscript after some minor revisions.

The abstract is overloaded with quantified results and is difficult to follow in the current form. Please shorten it and provide only the main quantified findings in fewer paragraphs.

[Figure]

Page 2, line 53 needs a reference.

Page 3, line 83 add months to the covered dataset period (e.g. Jan 2002 - Dec 2016).

Page 4, line 07 and page 6 lines 68-69 are there any additional uncertainty information (e.g. for hourly MFRSR data) about the angular response in order to exclude data for SZA>80 degrees?

Page 4, lines 20 and 28 provide a sentence explaining to the basics the Langley plot method with a relevant reference and mention the additional AOD uncertainty.

Maybe Sections 3 and 4 can be under the same Section with potential title "Global and diffuse PAR irradiances". Then, the current section 3 would be 3.1 under the title " All sky conditions" and the current section 4 would be 3.2 under the title "Cloud-free conditions", while the sub-sections will be renamed accordingly (3.1.1, 3.2.1 and 3.2.2 respectively).

Section 5 is recommended to be renamed as "Sensitivity analysis"

Add grid lines in Fig. 1.

link of the long term series with a discussion and details of the calibration series results and uncertainty

Since Lampedusa is severely affected by dust intrusions it would be important to discuss also aerosol attenuation in addition to Cloud attenuation. If the paper is focusing only on cloud effects this have to be mentioned clearly in the discussion or/and in the title. If there will be another manuscript describing the aerosol effects then this has to be mentioned too. Otherwise you can dedicate at least a paragraph on the issue including results of a number of publications on aerosol/dust effects on solar radiation at the site, that authors of this manuscript have investigated in depth in the past.

I would point out more in the conclusions the significance of this study (and PAR and diffuse PAR series, seasonal analusis e.t.c.) for agricultural related appications.

happy holidays and wishes for a happy new year, to all authors and the editor.

---

## Referee Comment (RC2) · Anonymous Referee #2 · 12 Mar 2018

This manuscript covers efficiently the topic of PAR from the point of view of statistics information and I suggest to be published in the frame of this special issue. I am quite satisfied with the work and the discussion done about the performance of the used instruments and my overall impression is that the authors have a well-established background about the topic. Eventhough ÎŹ could accept the manuscript as it is, I would like to make a few suggestions to the authors in order to improve it. Page 2, first paragraph (lines 46-49) and second paragraph (lines 50-53): References are needed Figure 1: As I can understand the real slope and the theoretical slope (linear fit) differ

more at low zenith angles where the PAR values are higher. Add an analysis about the dependence of the ratio PAR to PARL versus solar zenith angle. Page 6, line 74: "This interannual variability is larger for the diffuse than for the global PAR". Quantify both variabilities and then conclude the above sentence. Page 6, lines 76-78: "while diffuse PAR displays a more articulated seasonal evolution, with a first maximum in April and a secondary in June". Give a physical explanation for the two maxima. Figure 11 is useless and complicated. It should be removed. figure 12 &13: Since you apply linear regression in order to find the dependence, I suggest to apply multilinear regression for gaining clearer results and safer coclusions.

In the future, I hope that the authors should publish a work expanding and combining the findings of the current manuscript with theoretical data (radiation transfer model's outputs). It will be easier to qualify and quantify the factors affecting PAR.

---

## Author Comment (AC1) · 9 Apr 2018

This is a very interesting study on PAR measurements.
There are not too many studies on PAR long term measurements and this study is presenting a lot of new and interesting aspects of global and diffuse par
I would suggest the publication of the manuscript after some minor revisions.

*We thank the reviewer for the positive comments.*

The abstract is overloaded with quantified results and is difficult to follow in the current form. Please shorten it and provide only the main quantified findings in fewer paragraphs.

*The abstract was shortened as suggested.*

Page 2, line 53 needs a reference.

*The reference was added.*

Page 3, line 83 add months to the covered dataset period (e.g. Jan 2002 - Dec 2016).

*The months were added.*

Page 4, line 07 and page 6 lines 68-69 are there any additional uncertainty information (e.g. for hourly MFRSR data) about the angular response in order to exclude data for SZA>80 degrees?

*The study by Mizoguchi et al. (2010) shows that above 80° the cosine error may be of the order of 50%. Although the estimated error on the global irradiance is expected to be smaller, it may still be significant (of the order of 15-20% on the hourly mean).*

Page 4, lines 20 and 28 provide a sentence explaining to the basics the Langley plot method with a relevant reference and mention the additional AOD uncertainty.

*A short description was added.*

Maybe Sections 3 and 4 can be under the same Section with potential title "Global and diffuse PAR irradiances". Then, the current section 3 would be 3.1 under the title " All sky conditions" and the current section 4 would be 3.2 under the title "Cloud-free conditions", while the sub-sections will be renamed accordingly (3.1.1, 3.2.1 and 3.2.2 respectively).

*The sections were rearranged as suggested.*

Section 5 is recommended to be renamed as "Sensitivity analysis"

*We do not agree wit this suggestion. Maybe only part of section 5 might fall under this title. We have preferred to leave the title "Cloud effects" to this section.*

Add grid lines in Fig. 1.

*Gridlines were added to figure 1.*

link of the long term series with a discussion and details of the calibration series results and uncertainty

*This comment is not clear.*

Since Lampedusa is severely affected by dust intrusions it would be important to discuss also aerosol attenuation in addition to Cloud attenuation. If the paper is focusing only on cloud effects this have to be mentioned clearly in the discussion or/and in the title. If there will be another manuscript describing the aerosol effects then this has to be mentioned too. Otherwise you can dedicate at least a paragraph on the issue including results of a number of publications on aerosol/dust effects on solar radiation at the site, that authors of this manuscript have investigated in depth in the past.

*We are preparing a second paper dedicated to the aerosol influence on PAR at Lampedusa. This was very shortly stated in the paper (page 14, lines 44-45); we have better addressed this aspect in the introduction and in the conclusions.*

I would point out more in the conclusions the significance of this study (and PAR and diffuse PAR series, seasonal analysis e.t.c.) for agricultural related applications.

*We have added a comment on this topic. At the same time, since measurements used in the paper are obtained in a marine environment, we have also emphasized the potential interest for the estimate of primary production and biological processes occurring in the Mediterranean.*

Happy holidays and wishes for a happy new year, to all authors and the editor.

*We thank the reviewer and wish the best for the current year.*

---

## Author Comment (AC2) · 9 Apr 2018

This manuscript covers efficiently the topic of PAR from the point of view of statistics information and I suggest to be published in the frame of this special issue. I am quite satisfied with the work and the discussion done about the performance of the used instruments and my overall impression is that the authors have a well-established background about the topic. Eventhough I could accept the manuscript as it is, I would like to make a few suggestions to the authors in order to improve it.

*We thank the reviewer for the positive comments and for the suggestions.*

Page 2, first paragraph (lines 46-49) and second paragraph (lines 50-53): References are needed

*References were added in the first and second paragraph of page 2.*

Figure 1: As I can understand the real slope and the theoretical slope (linear fit) differ more at low zenith angles where the PAR values are higher. Add an analysis about the dependence of the ratio PAR to PARL versus solar zenith angle.

*We have checked that the linear relationship holds for different values of the solar zenith angle. There is some change in the slope with the selected solar zenith angle intervals; the differences are small (the slope changes between 0.99 and 1.07 when different ranges of solar zenith angles are considered) and the overall uncertainty remains within the stated estimates. It must also be pointed out that part of the large spread found for values of PAR above 50 W m$^{-2}$ is due to the role of small time differences between the two observations during cloudy periods. We have redrawn figure 1 removing the fit, which is not used in the analysis and is not discussed in the text.*

Page 6, line 74: "This interannual variability is larger for the diffuse than for the global PAR". Quantify both variabilities and then conclude the above sentence.

*The interannual variability of the annual mean global and diffuse PAR was quantified and discussed.*

Page 6, lines 76-78: "while diffuse PAR displays a more articulated seasonal evolution, with a first maximum in April and a secondary in June". Give a physical explanation for the two maxima.

*This effect is primarily due to the aerosol seasonal evolution. A sentence was added in the text.*

Figure 11 is useless and complicated. It should be removed.

*We have preferred to leave figure 11. Although it is complicated, is the basis for the discussion of several events which characterize the interannual variability of the cloud radiative effects. We believe that this discussion is useful to interpret the dataset, and would not be supported without the figure.*

figure 12 &13: Since you apply linear regression in order to find the dependence, I suggest to apply multilinear regression for gaining clearer results and safer conclusions.

*We thank the reviewer for the suggestion. We have applied a multilinear regression with respect to the three cloud properties. We have replaced figure 12 and the associated discussions with the results of the new analysis.*

In the future, I hope that the authors should publish a work expanding and combining the findings of the current manuscript with theoretical data (radiation transfer model's outputs). It will be easier to qualify and quantify the factors affecting PAR.

*We thank the reviewer for the suggestion. We are now writing a second paper dealing with the effects of aerosol, based on the cloud-free PAR and aerosol optical properties. The analysis incorporating radiation transfer modelling will be the object of future studies.*